# Infectivity and Transmissibility of Acute Hepatopancreatic Necrosis Disease Associated *Vibrio parahaemolyticus* in Frozen Shrimp Archived at −80 °C

Chorong Lee [1,†], Hye Jin Jeon [1,†], Bum Keun Kim [1], Seong-Kyoon Choi [2,3], Sumi Kim [4], Gwang Il Jang [4], Ji Hyung Kim [5,*] and Jee Eun Han [1,*]

1   College of Veterinary Medicine, Kyungpook National University, Daegu 41566, Korea;
    crlee@jejunu.ac.kr (C.L.); jhj1125@knu.ac.kr (H.J.J.); aam_kim@knu.ac.kr (B.K.K.)
2   Core Protein Resources Center, DGIST, Daegu 42988, Korea; cskbest@dgist.ac.kr
3   Division of Biotechnology, DGIST, Daegu 42988, Korea
4   Aquatic Disease Control Division, National Fishery Products Quality Management Service (NFQS),
    Busan 46083, Korea; sumikim@korea.kr (S.K.); gijang2@korea.kr (G.I.J.)
5   Department of Food Science and Biotechnology, Gachon University, Seongnam 13120, Korea
*   Correspondence: kzh81@gachon.ac.kr (J.H.K.); jehan@knu.ac.kr (J.E.H.)
†   These authors contributed equally to this work.

**Abstract:** Acute hepatopancreatic necrosis disease (AHPND) caused by *Vibrio parahaemolyticus* ($Vp_{AHPND}$) has been reported in commodity shrimp, but the potential risk of its global spread via frozen shrimp in the shrimp trade is yet to be fully explored. We hypothesized that frozen shrimp with AHPND could be a source of $Vp_{AHPND}$ transmission; thus, the infectivity of frozen shrimp with AHPND was evaluated using a shrimp bioassay. To prepare infected frozen shrimp, 12 *Penaeus vannamei* (average weight, 2 g) were exposed to $Vp_{AHPND}$ by immersion in water with a $Vp_{AHPND}$ concentration of $1.55 \times 10^7$ CFU mL$^{-1}$; once dead, the shrimp were stored at −80 °C for further analysis. After two weeks, a PCR assay was used to confirm AHPND positivity in frozen shrimp ($n = 2$), and $Vp_{AHPND}$ was reisolated from the hepatopancreases of these shrimp. For the infectivity test, 10 *P. vannamei* (average weight, 4 g) were fed with the hepatopancreases of $Vp_{AHPND}$-infected frozen shrimp ($n = 10$). After feeding, 70% of the shrimp died within 118 h, and the presence of $Vp_{AHPND}$ was confirmed using a PCR assay and histopathology examination; moreover, $Vp_{AHPND}$ was successfully reisolated from the hepatopancreases of the dead shrimp. We are the first to evaluate the potential transmissibility of $Vp_{AHPND}$ in frozen shrimp, and our results suggest that frozen shrimp with AHPND are a potential source of disease spreading between countries during international trade.

**Keywords:** bacteria; frozen; import; shrimp; transmission; *Vibrio*





## 1. Introduction

Loss of shrimp production due to infectious diseases is a major problem in shrimp aquaculture [1,2]. Nevertheless, millions of tons of frozen or processed commodity shrimp are traded internationally, despite the prevalence of disease in shrimp-producing areas that could be spread to other countries [3].

Acute hepatopancreatic necrosis disease (AHPND) is one of the most pathogenic and devastating bacterial diseases that affect global shrimp aquaculture. The virulence factor of AHPND is a binary toxin that is homologous with the *Photorhabdus* insect-related (*pir*) toxin; the toxin genes (*pir*A and *pir*B) are located in a large plasmid (69–73 kb) in *Vibrio parahaemolyticus* strains [4–6]. AHPND, which has been listed as a notifiable disease by the World Organization for Animal Health [7], has caused significant economic losses in the global shrimp aquaculture industry [8]. The pathogen can be detected in various sample types, although mainly in the hepatopancreas of infected shrimp; feces

and pond water are also sources of disease transmission on shrimp farms [9]. The possible transmission of AHPND-associated *V. parahaemolyticus* ($Vp_{AHPND}$) via the transboundary movement of infected broodstock or postlarvae has been reported [10], and live feeds, such as *Artemia* spp., rotifers, and polychaetes, are also known to be carriers and spreaders in farm environments [11,12].

In general, commodity shrimp are imported as frozen stocks and thawed before sale [13]. Freezing has been considered an efficient method for reducing the potential risk of seafood-borne zoonotic pathogen transmission, and frozen shrimp with AHPND have been considered noninfectious; therefore, such shrimp have not been a focus of the global industry, especially in relation to quarantine and disease surveillance [14–16]. Nevertheless, the occurrence of zoonotic bacteria (e.g., *Aeromonas* spp. and methicillin-resistant *Staphylococcus aureus*) in frozen shrimp has been reported [13,17,18]. Recently, we provided evidence that frozen commodity Pacific white shrimp (*Penaeus vannamei*) imported into Korea can be contaminated (or infected) with $Vp_{AHPND}$ [9]. However, the infectivity of $Vp_{AHPND}$ in frozen shrimp with AHPND has yet to be examined. Therefore, we investigated the infectivity of $Vp_{AHPND}$ in frozen shrimp as well as its transmissibility to live Pacific white shrimp.

## 2. Materials and Methods

### 2.1. Preparation of Frozen Shrimp with AHPND

For the $Vp_{AHPND}$ bacteria culture, the $Vp_{AHPND}$ strain (13-028/A3) [14] was cultured overnight at 28 °C in tryptic soy broth (Difco, Franklin Lakes, NJ, USA) containing 2.5% NaCl (hereafter referred to as TSB+) with shaking at 200 rpm. To prepare frozen shrimp with AHPND, specific-pathogen-free Pacific white shrimp (*n* = 12, average weight: 2 g) were purchased at a local shrimp farm (Tamla Shrimp, Jeju, Korea) and divided into two 12-L tanks. Water quality was maintained within a standard range for Pacific white shrimp as follows: temperature, 28–30 °C; pH, 7–8; and salinity, 25–26 ppt. In shrimp ponds, contaminated water is a primary source of AHPND infection, and we used the immersion method for the infection. The shrimp were inoculated with $Vp_{AHPND}$ via immersion in water with a $Vp_{AHPND}$ concentration of $1.55 \times 10^7$ CFU mL$^{-1}$. After immersion for 18 h, all shrimp (*n* = 12) died. Subsequently, frozen shrimp samples (*n* = 2) and hepatopancreas samples, which were aseptically collected from the remaining dead shrimp (*n* = 10), were stored at −80 °C until use. For a negative control, experimental shrimp (*n* = 20) that did not challenge with $Vp_{AHPND}$ were maintained in a 96 L acrylic tank and fed twice a day (5% of body weight) for three weeks. In those shrimp we confirmed that AHPND was negative by duplex PCR [5] and further used them for the $Vp_{AHPND}$ infectivity test (Section 2.3). The primer sets (*Vp*PirA–284F/R and *Vp*PirB–392F/R) were designed based on the virulence plasmid pVPA3–1 (GenBank No. KM067908). For detection of $Vp_{AHPND}$, PCR was performed as follows: 1 cycle of initial denaturation at 94 °C for 3 min, followed by 35 cycles of 94 °C for 30 s, 60 °C for 30 s, and 72 °C for 30 s, and a final extension at 72 °C for 7 min.

### 2.2. AHPND Polymerase Chain Reaction (PCR) Assay and $Vp_{AHPND}$ Bacterial Isolation from Frozen Shrimp

After being frozen for two weeks, the frozen shrimp samples (*n* = 2) were thawed at room temperature for 5 min. The hepatopancreases (average weight: 80 mg) were then collected under aseptic conditions from each shrimp. To detect $Vp_{AHPND}$, a portion of the frozen hepatopancreas (average weight: 30 mg) from each shrimp was pooled and used for DNA extraction, which was achieved using a DNeasy Blood & Tissue Kit (Qiagen, Hilden, Germany) according to the manufacturer's instructions. Extracted DNA was amplified using a duplex PCR assay [5]. The positive PCR amplicons were sequenced by Bioneer Inc. (Daejeon, Korea).

To isolate $Vp_{AHPND}$, the remaining portion of the frozen hepatopancreases (average weight: 50 mg) was homogenized and streaked on thiosulfate citrate bile salts sucrose

(TCBS) agar (Difco, Franklin Lakes, NJ, USA) using a wooden stick. Five representative green colonies were randomly chosen from the TCBS plates and inoculated in 20 mL of TSB+. After overnight incubation at 28 °C with shaking at 200 rpm, the enriched bacterial broth (1 μL) was used directly as a template for the duplex PCR assay [5]. The positive PCR amplicons were then sequenced by Bioneer Inc. (Daejeon, Korea).

*2.3. Infectivity Test for $Vp_{AHPND}$ in Frozen Shrimp*

Before the $Vp_{AHPND}$ infectivity test, three experimental shrimp were randomly selected and confirmed that the tested shrimp (hepatopancreas) did not carry other pathogens (white spot syndrome virus, infection hypodermal and hematopoietic necrosis virus and *Enterocytozoon hepatopenaei*) by PCR as described previously [19–21].

For the $Vp_{AHPND}$ infectivity test in frozen shrimp, Pacific white shrimp ($n$ = 20, average weight: 4 g) were divided into two groups (group 1: $Vp_{AHPND}$ challenged; group 2: no challenge) containing 10 shrimp each. The shrimp in each group were subdivided into duplicate groups of five shrimp in each 22 L tank. The infectivity test was conducted by feeding frozen hepatopancreas tissue because it was difficult to produce a large amount of contaminated water from frozen shrimp. We fed a total of 10 hepatopancreases of $Vp_{AHPND}$-infected frozen shrimp to group 1 shrimp ($n$ = 10). Thus, one shrimp was fed with one hepatopancreas (average weight of 80 mg) once. We also checked that all the shrimp in the tank consume the supplied hepatopancreas.

Experimental shrimp in group 1 ($Vp_{AHPND}$ challenged; $n$ = 10) were immediately fed with the frozen hepatopancreases ($n$ = 10), prepared as described in Section 2.1. The negative control shrimp in group 2 (no challenge; $n$ = 10) were not exposed to $Vp_{AHPND}$; they were fed with commercial shrimp feed at 5% of shrimp biomass. Shrimp survival was monitored and recorded for 224 h after feeding.

In this study, shrimp mortality started 46 h after feeding the frozen $Vp_{AHPND}$-infected hepatopancreas, and so shrimp was monitored every 24 h (46, 70, 94, 142, 166, 214 and 224 h) from that time point. Moribund shrimp samples ($n$ = 2, 46 h) were fixed in Davidson's AFA (alcohol, formaldehyde, acetic acid) fixative in preparation for histopathology analysis, and the hepatopancreases of dead shrimp samples ($n$ = 5) were used for $Vp_{AHPND}$ reisolation, which was confirmed using a PCR assay according to the method described above [5]. For the histopathological examination, hepatopancreas tissue sections were prepared using a standard method [22]. After staining with hematoxylin and eosin, the sections were analyzed using light microscopy.

## 3. Results

*3.1. Isolation of $Vp_{AHPND}$ from Frozen Shrimp*

After two weeks, frozen shrimp ($n$ = 2) stored at −80 °C, were confirmed to be positive for $Vp_{AHPND}$ via a PCR assay (Figure 1). More than 100 green colonies, i.e., presumptive *V. parahaemolyticus* isolates, were obtained from the frozen hepatopancreases after 48 h incubation at 28 °C on TCBS plates. According to colony PCR and sequencing analysis, the presence of $Vp_{AHPND}$ was confirmed in two selected single green colonies (Figure 1). The nucleotide sequences showed 100% identities to the known sequences of the *pir*A and *pir*B (accession no. KM067908). In addition, as a result of checking the hepatopancreas streaking plate after incubation, there were no colonies other than *Vibrio*.

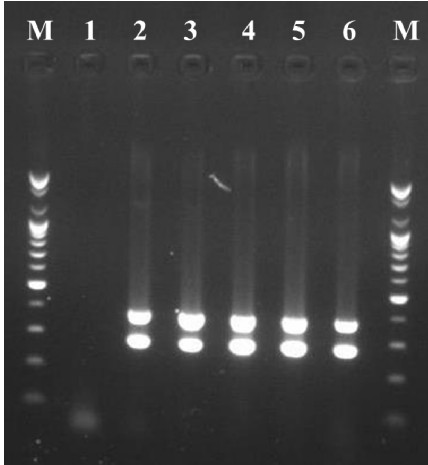

**Figure 1.** Duplex PCR assay was conducted to identify $Vp_{AHPND}$. Lane M: 100-bp DNA ladder; lane 1: negative control; lane 2: $Vp_{AHPND}$ positive control (from a shrimp farm in Vietnam); lane 3: hepatopancreas of shrimp frozen at −80 °C for two weeks; lanes 4 and 5: $Vp_{AHPND}$ isolates from frozen hepatopancreas; lane 6: dead shrimp after feeding on frozen hepatopancreas. Representative samples of each analysis are shown.

### 3.2. Infectivity of $Vp_{AHPND}$ in Frozen Shrimp

A laboratory shrimp bioassay was performed to determine the infectivity of frozen shrimp with AHPND. After feeding on frozen hepatopancreases that had been stored at −80 °C for two weeks, all experimental shrimp in group 1 ($Vp_{AHPND}$ challenged, $n = 10$) showed typical clinical symptoms of AHPND including pale hepatopancreases, softshells, and abnormal swimming behavior; in contrast, all shrimp in group 2 (no challenge, $n = 10$) showed normal appearance and behavior.

On the last day of the experiment, the experimental shrimp in group 1 ($Vp_{AHPND}$ challenged) had a 30% survival rate (Figure 2). Although the live shrimp showed typical AHPND symptoms at 118 h, those shrimp became healthy and showed normal external appearances and feeding behaviors on the termination day, shrimp. Duplex PCR analysis of the dead shrimp confirmed the presence of the toxin genes (*pir*A and *pir*B) (Figure 1), and $Vp_{AHPND}$ was reisolated from these shrimp ($n = 11$) (data not shown). In contrast, the survival rate in group 2 (no challenge) was 100% and the presence of toxin genes or $Vp_{AHPND}$ was not detected.

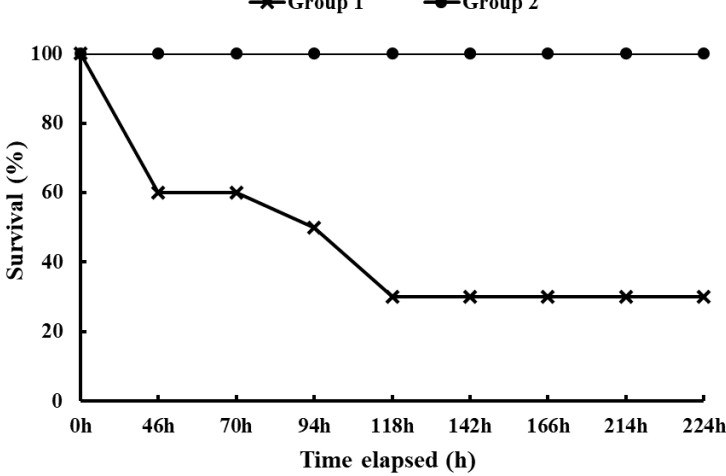

**Figure 2.** This Survival rate of Pacific white shrimp (*P. vannamei*) in an infectivity test involving feeding on frozen hepatopancreases infected with AHPND (group 1: $Vp_{AHPND}$ challenged; group 2: no challenge).

For histopathological examination, the moribund shrimp in group 1 ($Vp_{AHPND}$ challenged) were collected at 46 h postinfection. Overall, their epithelial cells were thinned and few vacuoles were observed in the hepatopancreas. Additionally, a typical AHPND infection region was observed in which the tubular structure of the cell was disrupted and combined with the surrounding debris (Figure 3A,B). However, abnormal tissue morphology was not observed in group 2 (no challenge) (Figure 3C,D).

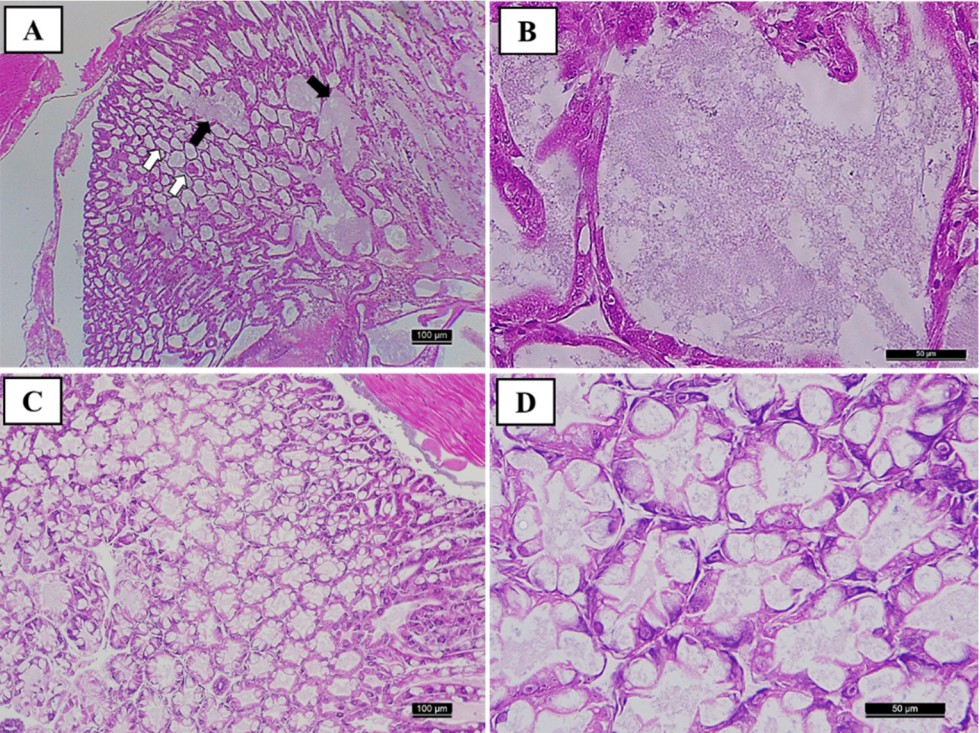

**Figure 3.** Histopathology examination using Davidson's AFA-fixed tissue sections from experimental shrimp fed on frozen hepatopancreas at 46-h postinfection. Histological signs in group 1 ($Vp_{AHPND}$ challenged) including tubular necrosis and cell debris are observed (**A,B**) (black arrows). Thin tubular epithelium is observed (white arrows). The hepatopancreases of group 2 (no challenge) had a normal appearance with normal cell functions (**C,D**).

## 4. Discussion

Shrimp is one of the most widely traded seafood products globally; thus, the global spread of disease through the international movement of frozen shrimp has become a serious problem [23–25]. Several studies have emphasized the risk of disease spreading among countries, especially for viral diseases such as infectious hypodermal and hematopoietic necrosis virus (IHHNV), white spot syndrome virus (WSSV), yellow head disease, and Taura syndrome virus [3,9,25–27].

In seafood, *V. parahaemolyticus* is of particular interest because it is consistently involved in seafood-borne disease outbreaks [28]. Interestingly, *V. parahaemolyticus* can be induced into a viable-but-nonculturable (VBNC) state under low temperature and salinity conditions; bacteria, including $Vp_{AHPND}$, are highly sensitive to freezing and refrigeration, both of which can lead to reductions in culture cells or non-detectable levels after several weeks [16,29]. Indeed, freezing at −18 to −24 °C reportedly kills *Vibrio* spp. due to its physical and chemical effects and possibly through induced genetic changes [30]. Additionally, the bacteria in shrimp are thought to become inactivated by freezing and thawing [31]. Moreover, frozen shrimp prepared and packaged for retail and wholesale stores or block frozen shrimp are considered unlikely to a pose threat to the international spread of disease [10].

However, several studies on the VBNC state of *V. parahaemolyticus* have reported the potential risks of revival and international transmission with elevated disease burden. First, bacteria in the VBNC state can retain their plasmids and virulence properties [32]. These characteristics are of great importance in relation to $Vp_{AHPND}$ because the *pir*A/B toxin genes are located in the plasmid; thus, plasmid retention in the VBNC cells of frozen shrimp with AHPND will increase the likelihood of disease spread and transmission. Second, *V. parahaemolyticus* VBNC cells can be resuscitated by increasing temperature; the resuscitation period is generally around two weeks after becoming unculturable [33,34]. Third, the possible presence of culturable VBNC cells from *V. parahaemolyticus* has been confirmed in frozen commodity shrimp that tested negative for *V. parahaemolyticus* using the ISO 21872-1 method [32]. Therefore, $Vp_{AHPND}$ cells in frozen shrimp with AHPND have the potential for resuscitation and consecutive infection.

Based on the aforementioned characteristics, we hypothesized that frozen shrimp with AHPND could be a source of $Vp_{AHPND}$ transmission, and we tested this hypothesis using a shrimp bioassay. We found that shrimp fed with $Vp_{AHPND}$-infected shrimp previously stored at $-80\,^{\circ}$C for two weeks became moribund and showed the typical clinical symptoms of AHPND at 46 h postinfection. Additionally, the $Vp_{AHPND}$ strain was successfully reisolated from shrimp frozen at $-80\,^{\circ}$C and dead shrimp. Thus, we demonstrated the infectivity of $Vp_{AHPND}$ in frozen shrimp with AHPND, which could therefore be a source of disease transmission between countries. This finding is inconsistent with the results of Thiamadee et al. [10], who suggested that low temperatures reduce *Vibrio* viability and that $Vp_{AHPND}$ cannot be transmitted via traded frozen shrimp. We agree that the riskiest activity for the geographical spread of $Vp_{AHPND}$ is the transboundary movement of living shrimp broodstock or their offspring from an area affected by $Vp_{AHPND}$ to an unaffected area for aquaculture [10]. However, our results also suggest that the uncontrolled movement of frozen shrimp with AHPND could increase the risk of spreading AHPND from an area with an outbreak to an unaffected area.

In Korea, IHHNV and WSSV have been detected in frozen commodity shrimp and crayfish imported from Vietnam, Indonesia, and China (the major exporters of cultured crustaceans), and the infectivity of these viruses as well as the risk of disease transmission into healthy shrimp have been examined [35,36]. Additionally, the *pir*A/B toxin genes of $Vp_{AHPND}$ were detected in frozen shrimp imported from Vietnam and Indonesia to Korea [9]. If frozen commodity shrimp with AHPND retain viable or resuscitable $Vp_{AHPND}$ that could infect other shrimp, quarantine of frozen shrimp will be important for the prevention of AHPND spread, particularly as those imported shrimp could lead to serious economic losses in the domestic shrimp farms of Korea. Therefore, to prevent AHPND transmission, frozen shrimp importers should be more aware of the risks and develop stricter quarantine policies. Moreover, it will be necessary to examine the infectivity of other important bacterial diseases in cultured shrimp in future research.

**Author Contributions:** Writing-original draft preparation, C.L. and H.J.J.; formal analysis, B.K.K.; methodology, S.-K.C.; investigation, S.K. and G.I.J.; writing—review and editing and supervision, J.H.K. and J.E.H. All authors have read and agreed to the published version of the manuscript.

**Funding:** This work was supported by the National Research Foundation of Korea (NRF) grants [NRF-2018R1C1B5086350, NRF2019R1C1C1006212, NRF-2020R1I1A2068827, and NRF2021R1I1A1A01040303], and DGIST R&D Program [2020010096] of the Ministry of Trade, Industry. This research was supported by the project titled "Development of quarantine & disease control program for aquatic life" funded by the National Fishery Products Quality Management Service. This research was also supported by Development of technology for biomaterialization of marine fisheries by-products of the Korea institute of Marine Science & Technology Promotion (KIMST) funded by the Ministry of Oceans and Fisheries (KIMST-20220128).

**Institutional Review Board Statement:** The animal study protocol was approved by the Ethics Committee of Kyungpook National University (KNU 2021-0066 and 29 April 2021).

**Informed Consent Statement:** Not applicable.

**Data Availability Statement:** Not applicable.

**Conflicts of Interest:** The authors declare no conflict of interest.

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
