# Peer review of "Infectivity and Transmissibility of Acute Hepatopancreatic Necrosis Disease Associated Vibrio parahaemolyticus in Frozen Shrimp Archived at −80 °C"

_fishes, doi:10.3390/fishes7030125_

Round 1
Reviewer 1 Report
This manuscript by Lee et al, entitled “Infectivity and transmissibility of acute hepatopancreatic necrosis disease associated Vibrio parahaemolyticus in frozen 3 shrimp archived at -80℃” explored the potential risk of bacterial pathogens/diseases, i.e., Acute hepatopancreatic necrosis disease (AHPND) being transmitted via the frozen storage chain of shrimp, especially shrimp carrying the strain of Vibrio parahaemolyticus that causes AHPND. In all, the study is of scientific interest and the work was well planned and executed. That said, there are a few things that need to be resolved before the manuscript is put in a better position for publication. Details of my comments are shown below.
- I am not sure why the authors decided to freeze their shrimp at -80℃ because most of the exported shrimp are not frozen at such a temperature. Mostly, -40℃ or higher temperatures are used. Could authors, therefore, justify the choice of the temperature used, if this is to give a true reflection of the danger posed by frozen shrimp carrying AHPND.
- To prepare the infected frozen shrimp, the authors used small shrimp (average weight of 2 g), while for the infectivity test, relatively large shrimp (average weight of 4 g) were used. Why were different shrimp used?
- Authors stated that “shrimp were fed with the hepatopancreases of VpAHPND -infected frozen shrimp” why were two different methods used, i.e., immersion for the initial infection of shrimp, and feeding for the infectivity testing?
- Could authors also elaborate on how the shrimp (average weight of 4 g) were fed with hepatopancreas, given their size?
- If residual pathogenic bacteria such as the one that causes AHPND is unavoidable and traces can still be found in frozen shrimp, could authors suggest alternative preservative methods or how the shrimp could be treated before freezing, albeit without loss of aesthetic qualities?
- Which gene(s) was amplified in the PCR used to confirm VpAHPND positivity because? Although the authors referenced a paper, I suggest the PCR conditions, primers, genes, etc. should be stated. Moreover, there are different genes used to confirm VpAHPND (please refer to the 2019 OIE - Manual of Diagnostic Tests for Aquatic Animals, https://www.oie.int/fileadmin/Home/eng/Health_standards/aahm/current/chapitre_ahpnd.pdf)
- Timing of the challenged experiments and sample collection. I am confused about the timing for sample collection. Why didn’t the authors use times such as 12, 24, 36, 48, etc.?. Please justify the timing
- In the survival analysis, I wonder why the shrimp didn’t die after 118 h? could the authors offer some explanation or speculate on this?
- Did the authors also analyze total bacterial abundance, because some opportunistic pathogens or coinfected pathogens could also survive the freezing and therefore pose a potential risk?
- Although the focus of the current study is on AHPND, it would have been good if the authors had also analyzed other common pathogens to see if they survived the freezing and therefore could also pose danger.
Reviewer 2 Report
The authors studied the infectivity of Vibrio parahaemolyticus in shrimp frozen at -80°C and its transmissibility to Pacific white shrimp. The presented manuscript broadens the current knowledge about acute hepatopancreatic necrosis disease (AHPND), that causes significant losses to the global shrimp industry.
Comments
Line 73. Information about feeding and quarantine of used shrimp should be added. Was a control group of shrimp in the experiment?
Line 86. The names of genes detected and all primers used in the duplex PCR method should be added.
Line 99. Information about quarantine of shrimp should be clarify.
Line 100-101. Shrimp was feeding with the frozen hepatopancreas which was positive for Vibrio parahaemolyticus. Have hepatopancreas used for feeding been tested for the presence of other pathogens or bacteria?
Line 117. Colony should be replaced by colony morphology. It should be clarified if the bacteria were isolated in the monoculture.
Line 118. If own sequence was compared with reference? It should be clarified.
Line 135. How many colonies were isolated from the shrimp? Was the different number compared to frozen shrimp described in line 115? Was the bacteria isolated in a monoculture? It should be clarified.
